# General Flow as Foundation Affordance for Scalable Robot Learning

**Chengbo Yuan, Chuan Wen, Tong Zhang, Yang Gao**[†]
Institute for Interdisciplinary Information Sciences, Tsinghua University
Shanghai Qi Zhi Institute
Shanghai Artificial Intelligence Laboratory
{ycb24, cwen20, zhangton20}@mails.tsinghua.edu.cn
gaoyangiiis@mail.tsinghua.edu.cn
[†]Corresponding Author

**Abstract:** We address the challenge of acquiring real-world manipulation skills with a scalable framework. We hold the belief that identifying an appropriate prediction target capable of leveraging large-scale datasets is crucial for achieving efficient and universal learning. Therefore, we propose to utilize 3D flow, which represents the future trajectories of 3D points on objects of interest, as an ideal prediction target. To exploit scalable data resources, we turn our attention to human videos. We develop, for the first time, a language-conditioned 3D flow prediction model directly from large-scale RGBD human video datasets. Our predicted flow offers actionable guidance, thus facilitating zero-shot skill transfer in real-world scenarios. We deploy our method with a policy based on closed-loop flow prediction. Remarkably, without any in-domain finetuning, our method achieves an impressive 81% success rate in zero-shot human-to-robot skill transfer, covering 18 tasks in 6 scenes. Our framework features the following benefits: (1) scalability: leveraging cross-embodiment data resources; (2) wide application: multiple object categories, including rigid, articulated, and soft bodies; (3) stable skill transfer: providing actionable guidance with a small inference domain-gap. Code, data, and supplementary materials are available https://general-flow.github.io/.

**Keywords:** Flow, Transferable Affordance, Human Video

## 1 Introduction

We aim to develop a new framework that enables scalable learning for robot manipulation. With more data and larger model training in the future, this framework has the potential to progressively enhance the capabilities of robots, i.e., the scaling law that has been observed in LLMs [1]. Inspired by the LLMs training paradigm [2], we believe that two key elements contribute to their strong generalization abilities: (1) a vast training dataset with a small inference domain gap, such as all texts from the internet in LLMs, and (2) a foundational prediction task, such as text-token prediction in LLMs. How can we translate these elements into robot learning?

Confronted with the challenges of collecting real-world robot data [3, 4], we pivot towards large-scale human video datasets. These data resources guarantee scalability and a small inference domain-gap (no simulation-to-reality problem), key ingredients for effective generalization. Moreover, human manipulation data provides a vast, real-world resource rich in diverse physics interactions and dynamic behaviors that closely align with robot manipulation. The next step is to identify a proper prediction target. We propose affordance for this role. Rooted in Gibson's theory [5], affordance concentrates on the actions associated with an object, remaining neutral to manipulators. This characteristic positions affordance as a cornerstone in humtn-to-robot skill transfer.

What affordance format will lead to a foundation prediction target that is universal for object categories and provides actionable guidance for robot manipulation? In this paper, we propose **General**

8th Conference on Robot Learning (CoRL 2024), Munich, Germany.

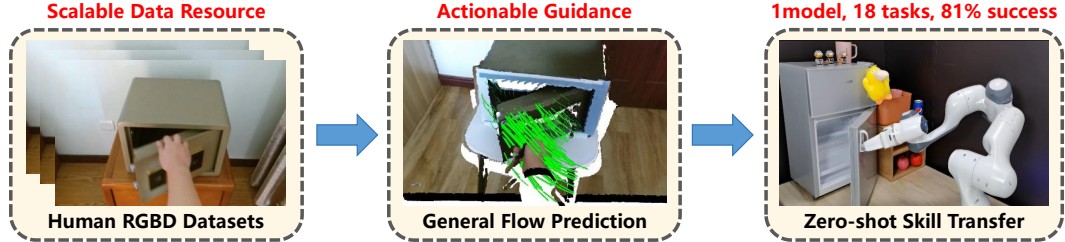

Figure 1: We propose General Flow as a foundational affordance. Our framework uses general flow affordance as a bridge representation for human-to-robot skill transfer. Trained solely on RGBD human video datasets, our system achieves an average success rate of 81% on 18 real-world robot manipulation tasks, highlighting a pathway for scalable robot learning.

**Flow as Foundation Affordance** (as shown in Figure 1) to achieve this goal. This affordance elucidates the future trajectories of 3D points on the object of interest. Take the task of 'open Safe' as an instance (in the middle of Figure 1): the general flow represents future positions of points on the safe. Then, a robot can gain a resilient motion primitive for the opening skill by following the door's flow.

Previous works [6, 7, 8, 9] have attempted to extract 3D flow representations from either simulation or real robot data. However, these approaches suffer from domain transfer gaps [6] or have limited scalability [7]. In contrast, general flow leverages real-world, scalable data resources, i.e., human videos, thereby eliminating both the sim-to-real domain gap and the dependence on burdensome robot data collection. The diversity of backgrounds, objects, and human behaviors in these videos also significantly enhances the robustness of real robot execution. We term our affordance "general flow" due to its capability for universal robot learning: **(1) scalability**: leveraging scalable human data resources; **(2) wide application**: multiple object categories, including rigid, articulated, and soft bodies. **(3) stable skill transfer**: providing actionable guidance with a small inference domain-gap, even sufficient for zero-shot execution.

In this paper, we first develop pipelines to extract 3D flow labels directly from RGBD human video datasets for model training. We find prediction of dense flow in real-world scene point clouds remains a challenge, primarily due to the variability of trajectory scales and the need to enhance robustness in zero-shot scenarios. To address these issues, we employ scale-aware strategies in the model aspect, complemented by augmentation techniques that focus on embodiment occlusion (human hand and robot arm) and query point sampling (3D points on objects of interest), thereby boosting zero-shot stability.

Implementing a straightforward heuristic policy derived from closed-loop flow prediction, we evaluate our approach on a Franka-Emika robot in a real-world setup. **Without any in-domain fine-tuning**, our system accomplishes stable zero-shot human-to-robot skill transfer. Fueled by the rich actionable guidance of general flow affordance, **with only one model**, our system notches **an impressive 81% average success** rate in 18 diverse tasks across 6 scenes, covering multiple categories of object types like rigid, articulated, and soft bodies. These findings highlight the transformative potential of general flow in spearheading scalable general robot learning.

## 2   Problem Formulation

The affordance [5] typically consist of functional grasp and subsequent motion [10, 11]. While robust functional grasping has been extensively explored [12, 13, 14], a general method for providing prior guidance for post-grasp movements remains a challenge, which is the primary focus of our paper.

Understanding the semantics and geometry of post-grasp motion for accurate predictions is complex. For example, in the "open Safe" task, the model must infer that the door's point trajectory rotates around an implicit axis, while points on the base remain static. Additionally, it must note that the

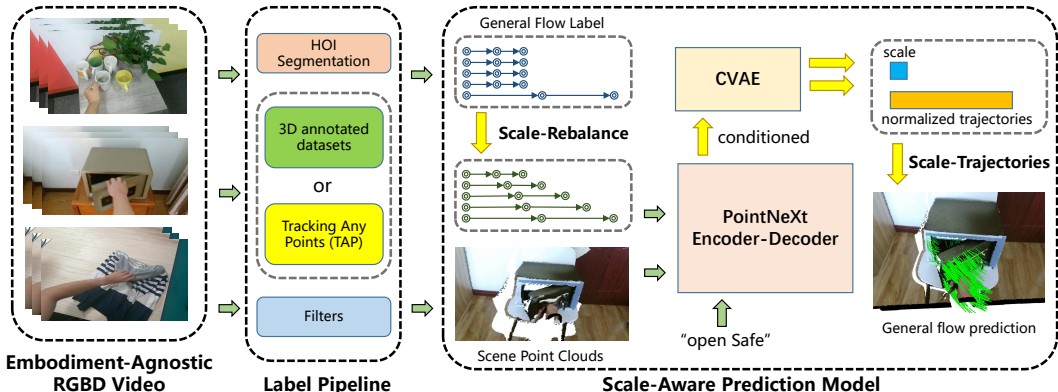

Figure 2: The framework of our prediction model. We build pipelines to extract general flow labels from both RGBD human video datasets. Then, multiple design elements are utilized to enhance the scale-awareness and robustness of the prediction model.

trajectories of closer points to the axis require a smaller predicted scale or length. To handle these, we introduce "general flow" as an affordance that provides **actionable guidance** for downstream manipulation:

**Definition of General Flow** Given a perception observation $S$ (from any embodiment) and a task instruction $I$, for $N_q$ 3D query points $Q \in R^{N_q \times 3}$ in space, the general flow $F \in R^{N_q \times T \times 3}$ represents the trajectories of these points over $T$ future timestamps.

**Detailed Formulation** In this work, we use point clouds from real-world RGBD camera streams as our perception state $S$. Our model processes natural language instructions $I$, scene point cloud features $P_s \in R^{N_s \times 6}$ (comprising $N_s$ points with XYZ + RGB attributes), and $N_q$ spatial query points $Q \in R^{N_q \times 3}$ (comprising $N_q$ points with XYZ attributes). The aim is to predict a trajectory set, or "flow", denoted as $F \in R^{N_q \times T \times 3}$. For the $i$-th query point $p^i \in R^3$, its trajectory is defined as $F^i \in R^{T \times 3}$, with the absolute position at time $t$ represented as $F_t^i \in R^3$ for $t = 1, 2, \cdots, T$. Initially, $F_0^i$ is set to the input position of the query point $p^i$.

We term our affordance "general flow" to emphasize its broad applicability across different embodiments (from any embodiment to any robot embodiment, e.g. from human to Franka-Emika) and object categories (e.g. rigid, articulated and soft bodies).

## 3 Embodiment-Agnostic Flow Prediction

We propose a framework for general flow prediction that is agnostic to specific embodiments, which is outlined in Figure 2. We first design pipelines to extract flow labels from RGBD human video datasets. To manage various scales of trajectories and account for real-world noise, we integrate key designs that enhance the model's scale-awareness and robustness in predictions. For the first time, we develop a language-conditioned 3D flow prediction model directly from large-scale RGBD human video datasets.

### 3.1 General Flow Label Acquisition

We extract general flow labels from two types of cross-embodiment datasets. For rigid and articulated objects, we utilize the HOI4D dataset [15] to train our general flow prediction model. This extensive RGBD video dataset includes 16 categories and 800 objects, encompassing 44.4 hours of recording. It provides comprehensive 3D labels, such as active object segmentation, 3D pose, and camera parameters. To further explore general flow in soft object manipulation, we collect RGBD videos for the "fold clothes" task using the RealSense D455 camera, comprising 6 types of clothes, 30 trajectories, and 605 extracted clips.

| Model | ADE↓ | FDE↓ | PM |
|---|---|---|---|
| ResNet18 | 7.54 | 10.71 | 13.2 |
| R3M(frozen) | 7.55 | 10.56 | 11.9 |
| R3M(finetune) | 7.54 | 10.69 | 11.9 |
| VAT-MART | 7.16 | 12.20 | 1.6 |
| VIT-B-224 | 6.81 | 9.48 | 86.6 |
| PointNeXt-B | 3.96 | 5.37 | 4.1 |
| PointNeXt-L | 3.83 | 5.16 | 15.6 |
| ScaleFlow-S | 3.74 | 5.01 | 0.9 |
| ScaleFlow-B | _3.58_ | _4.77_ | 5.6 |
| ScaleFlow-L | **3.55** | **4.70** | 17.1 |

Table 1: Results of general flow prediction on the test set. The unit of ADE and FDE is "cm". PM refers to ParaMeters with unit million.

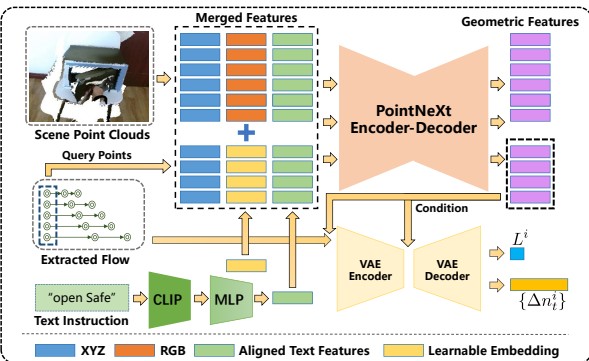

Figure 3: Design architecture of our model. The model employs a CLIP encoder to convert instructions into semantic features and utilizes a PointNeXt backbone along with a conditional VAE to capture the multimodality of different action trajectories.

**From HOI4D Datasets[15]:** Utilizing the detailed 3D labels from these datasets, we first randomly sample points within the active object and then calculate its future position using ground-truth pose and camera parameters.

**From Collected RGBD Videos:** We first perform Human-Object-Interaction (HOI) segmentation [16, 17] to obtain the active object mask. Points are then sampled within the mask, and the future 2D trajectory is tracked using the Tracking Any Point (TAP) models [18]. The 3D label of the general flow is computed through back projection in both the spatial and temporal dimensions.

To reduce the effect of noise in the annotations and the pipeline, multiple filtering techniques are employed. Additionally, we retain the hand mask for potential use in training augmentations. More details are shown in Appendix B.

### 3.2 Scale-Aware Flow Prediction

We observe enhanced performance when predicting relative displacements rather than absolute positions. I.e., we predict $\Delta p_t^i = F_t^i - F_{t-1}^i$ instead of $F_t^i$. The illustration of the model architecture is shown in Figure 3. Following the description in Section 2, we first use CLIP [19] encoder to convert instructions into semantic features. Then their dimensions are reduced via MLPs (to $d_I$ dimensions) to align with point features. We first concatenate aligned text features with point cloud RGB+XYZ features, and then concatenate the features of scene points and query points, forming merged point cloud features $P_M \in R^{(N_s+N_q) \times (3+3+d_I)}$. The merged features are processed through a PointNeXt [20] backbone with a segmentation head to extract geometric features. Since humans may perform different action trajectories for the same task and scene, we utilize a VAE [21, 22] to capture this multimodal property. The features of query points serve as condition variables for a conditional VAE, generating the final predictions. We leave more details of architecture in Appendix C.1.

A primary challenge in real-world flow prediction is the significant variance in trajectory lengths across different query points within a task. For instance, in the "open Safe" task, the trajectories of points on the door are substantially longer than those on the safe body. To address this, we apply the Total Length Normalization (TLN) to uniformly rescale trajectories. We define trajectory scale for each query point $p^i$ as $L^i = \sum_{t=1}^{T} ||\Delta p_t^i||$. For the original prediction target $\{\Delta p_t^i \mid t = 1...T\}$, we define the normalized target $\{\Delta n_t^i\}$ as:

$$\Delta n_t^i = \frac{\Delta p_t^i}{L^i} \tag{1}$$

Then the VAE predicts the scale $L^i$ and normalized trajectory $\Delta n_t^i$ separately. Our ablation study demonstrates that TLN yields the best performance compared with other normalization methods (Appendix F.1).

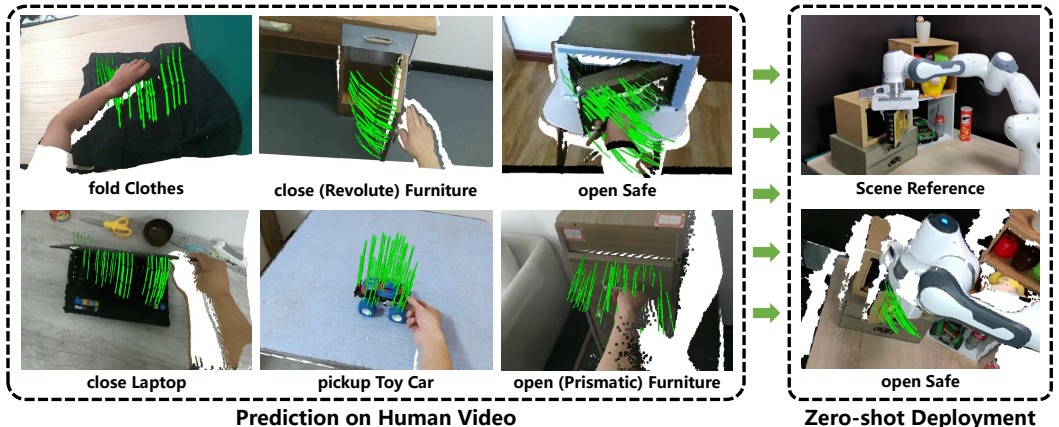

Figure 4: Visualization of general flow prediction. With only trained on human video datasets, our model could predict general flow robustly in zero-shot robot deployment scene.

## 3.3 Training Flow Prediction Model

The final loss function comprises trajectory prediction loss $\mathcal{L}_{traj} = \frac{1}{N_q} \sum_{i,t} ||\Delta \hat{n}_t^i - \Delta n_t^i||^2$, scale regression loss $\mathcal{L}_{scale} = \frac{1}{N_q} \sum_{i,t} ||\hat{L}^i - L^i||^2$, and KL-divergence loss $\mathcal{L}_{KL}$ of the VAE[21]. To minimize cumulative error, we also incorporate an MSE loss $\mathcal{L}_{acc} = \frac{1}{N_q} \sum_{i,t} ||\hat{F}_t^i - F_t^i||^2$ for the recovered accumulative shift. Thus, the total loss is expressed as:

$$\mathcal{L} = \mathcal{L}_{traj} + \beta_1 \mathcal{L}_{scale} + \beta_2 \mathcal{L}_{KL} + \beta_3 \mathcal{L}_{acc} \tag{2}$$

In light of the complex environmental challenges encountered in zero-shot real-world deployments in later sections, we propose two technical augmentations to boost zero-shot generalization robustness. (1) Hand Mask Augmentation (HMA): to enhance resilience to embodiment occlusions, we drop out the hand points in the input scene point clouds with some probabilities. (2) Query Points Sampling (QPS): to adapt to varying query point distributions required by different applications, we augment training by switching query point distributions. For more details of the augmentations and training procedures, please refer to Appendix C.2, C.3. Additionally, an ablation study is presented to verify the effectiveness of all design elements in Appendix F.

## 3.4 Evaluation

The collected general flow dataset is divided into training, validation, and test sets in an 80%, 10%, 10% ratio, with no identical object instances across sets. We adopt three types of models as our baselines (demonstrated in Appendix D). (1) 2D model: ResNet[23], ViT[24], R3M[25]. (2) Point clouds affordance model: VAT-MART[10]. (3) 3D model: PointNeXt[20], which is an improved version of the PointNet++[26] used in Flowbot3D[6] and ToolFlowNet[7]. We refer to our model as "**ScaleFlow**" in subsequent discussions. We trained multiple versions of the PointNeXt-based models, each with different PointNeXt backbone sizes.

We use 3D Average Displacement Error (ADE) and Final Displacement Error (FDE) in centimeters [27, 28] as evaluation metrics. For VAE-dependent models, metrics are averaged over 10 samplings. The results in Table 1 demonstrate that our model achieves superior performance on all metrics, even with fewer parameters. We show the visualization of flow prediction in Figure 4. Moreover, our system, trained at scale, demonstrates **notable emergent properties** such as robustness to hand occlusion, language-driven semantic control, resilience to label noise, and scale spatial adaption. We demonstrate these properties in Appendix E.1 and validate them in real-robot experiment in Appendix E.2.

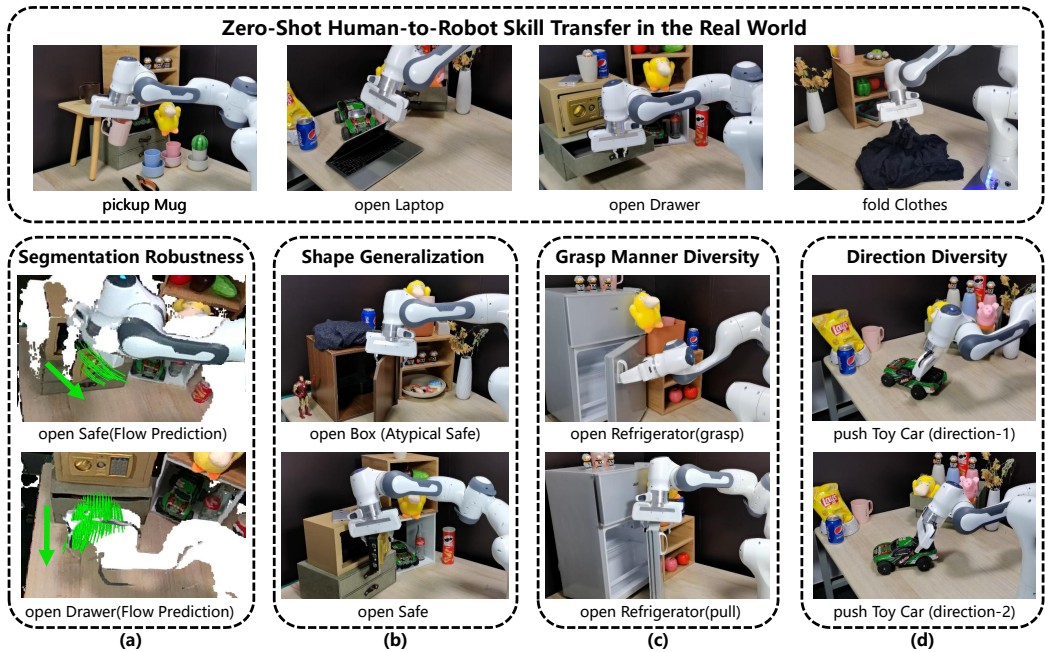

Figure 5: We achieve stable zero-shot human-to-robot skill transfer in the real world, encompassing 18 tasks with rigid, articulated, and soft objects across 6 scenes.

## 4  Zero-shot Real World Manipulation

In this section, our goal is to demonstrate the efficacy of general flow as an actionable guidance [29] for downstream robot manipulation. We choose one of the most challenging scenarios: **zero-shot human-to-robot skill transfer in the real world**. Utilizing **only one prediction model for all tasks** paired with a straightforward heuristic policy derived from closed-loop flow prediction, we achieve an impressive 81% average success rate. This success spans categories including rigid, articulated, and soft objects, and covers 18 tasks across 6 distinct scenes.

### 4.1  Heuristic Policy with General Flow

Here we present our heuristic policy based on close-loop flow prediction (Algorithm 1 in Appendix). We use a RealSense D455 RGBD camera positioned behind the Franka-Emika Arm to capture an ego-view stream. The robot's static base during manipulation acts as a reference for the FastSAM model [17] to segment the robot. More prompt points could also be employed via a designed GUI. Post-segmentation, we reconstruct 3D scene point clouds and select query points near the gripper. These query points, treated as a miniature rigid body, together with the scene point clouds and the text instruction, are fed into our to predict the general flow.

The next step involves solving the corresponding transformation, an ICP (Iterative Closest Point) problem, based on the general flow prediction. We employ the SVD (Singular Value Decomposition) algorithm [30] to align the robot arm's end-effector transformation with the predicted flow. The choice of SVD is due to its robustness in handling noise and outliers, as well as its computational efficiency. Once the aligned transformation is computed, we use the Deoxys library [31] as an impedance controller in the operation space to execute the transformation accurately. We leave more details of deployment system and policy derivation in Appendix G.3 and Appendix G.4

### 4.2  Real World Experiment Setting

For real-world experiments (Figure 5 and Appendix G.1), we select 8 objects (covering rigid, articulated and soft bodies) across 6 scenes, encompassing 18 manipulation tasks. The rigid category includes Mug and Toy Car. Articulated objects are Safe, Box (which can be approximated as an

| Object | Action-1 | SR-1 | Action-2 | SR-2 | Action-3 | SR-3 |
|---|---|---|---|---|---|---|
| Mug | pickup | 10/10 | putdown | 9/10 | - | - |
| Toy Car | pickup | 10/10 | putdown | 10/10 | push | 5/10 |
| Clothes | fold | 8/10 | - | - | - | - |
| Safe | open | 9/10 | close | 10/10 | - | - |
| Box | open | 10/10 | close | 10/10 | - | - |
| Drawer | open (pull) | 4/10 | open (grasp) | 3/10 | close | 10/10 |
| Refrigerator | open (pull) | 7/10 | open (grasp) | 9/10 | close | 10/10 |
| Laptop | open | 5/10 | close | 7/10 | - | - |
| **Average Success Rate** | | | **81% (146 / 180)** | | | |

Table 2: Result of real-world manipulation with **one model for all tasks**. "SR" refers to the success rate. For the "open" task of "Storage Furniture", "pull" means execution with an opened gripper in a pulling manner, while "grasp" is with a closed gripper on the handle.

atypical design of Safe), Laptop, Refrigerator, and Drawer, while the soft category includes Clothes. We perform "pickup" and "putdown" actions for rigid objects, with an additional "push" action for the Toy Car. Articulated objects undergo "open" and "close" tasks, and the soft object is subjected to "fold" action. (See Figure 10, 11 in Appendix G.1 for visualization). All object categories (except for Box) and manipulation tasks appear in the training data. Except for the "Toy Car" and "Safe", all other objects are novel instances for model training. To demonstrate the generalization ability of our model, we randomly build the environment backgrounds, ensuring that all evaluation settings are unseen during training.

As general flow affordance guides post-grasp motion, we manually position the robotic arm for task initiation. This can be replaced with automatic methods, as demonstrated in Ko et al. [32]. For storage furniture with handles, we evaluate the performance of both opened and closed grippers. Each task undergoes 10 trials, and the success rates are recorded. Discussions about the quantitative success criteria of each task can be found in Appendix G.1. We also discuss the real-world baseline model [11] in Appendix G.2.

### 4.3 Results and Analysis

For result analysis, we focus on the following keywords and ask the following questions:

- **Transfer Ability:** Does general flow facilitate stable zero-shot human-to-robot transfer?

- **Segmentation Error:** How robust is the system against robot segmentation errors?

- **Novel Shape:** Can this model generalize to the shapes of new categories that are significantly different from the training instances?

- **Grasp Manner:** Is this object-centric system robust to variations in grasp manner?

- **Diverse Setting:** How well can general flow adapt to diverse scenes and setting?

Figure 5 presents a comprehensive overview of our analysis distribution. Following this, we delve into a detailed quantitative and qualitative examination to address these questions.

**Stable Zero-Shot Skill Transfer** Our results, presented in Table 2, demonstrate that using general flow as a bridge enables our framework to achieve stable zero-shot human-to-robot skill transfer. An impressive 81% success rate in such challenging settings underscores the strong transfer ability of general flow in cross-embodiment robot learning. To our knowledge, this is the first flow-based work to reach such a level of zero-shot transfer performance in real-world experiments. For tasks with success rates below 60%, we meticulously analyze the reasons and propose feasible future solutions in Appendix G.6. We also discuss the inference latency of our system in Appendix G.5.

**Robustness to Segmentation Error** Our findings reveal that random hand mask augmentation during training significantly enhances the model's robustness to errors in the segmentation maps of

FastSAM [17]. Figure 5(a) illustrates this advantage with two examples. Notably, even with almost failed robot segmentation (as in the "open safe" task), our method still predicts meaningful flow to facilitate task completion in a closed-loop manner.

**Generalization to Novel Shape** To probe the boundary of general flow's generalization capabilities, we experiment with a "Box" category, which can be approximated as an atypical design of "Safe". For comparison, we also test a conventional "Safe". Figure 5(b) presents these instances. Surprisingly, the success rate for manipulating "Box" is even higher than that of the ordinary one (100% vs. 90% for the "open" task, Table 2), attributed to the structure of "Box" allowing more trajectory deviation without losing gripper on the door. This underscores the strong generalization capacity of general flow methods.

**Robustness to Grasp Position and Manner** As general flow is an embodiment-agnostic and object-centric method, it is expected to be resilient to variations in gripper position and grasp manner. To test this, we conduct manipulations using two storage pieces (a Refrigerator and a Drawer), leveraging their handles for different grasp methods. Figure 5(c) displays these different execution manners. Our model successfully completes tasks regardless of the gripper's state.

**Handling Diverse Scenes and Directions** We examine the extent to which general flow can handle changes in scene and direction. We distribute our tasks across six diverse scenes and perform scene-based prediction, eliminating the need for clean segmentation of the manipulated object. We also vary the direction of movable objects during our experiments. Figure 5(d) showcases the most challenging example in this regard. We find that our heuristic policy successfully pushes a Toy Car in different directions to a certain extent.

## 5 Related Work

**Embodiment-Agnostic Framework for Scalable Robot Learning** We discuss more related works of real-world general robot learning in Appendix H. To leverage large-scale, cross-embodiment data resources [33, 34, 35, 15, 36], multiple embodiment-agnostic frameworks [37] are proposed for robot learning. Prior works [25, 38, 39, 40] employ large-scale visual pre-training to develop embodied-aware pretrained representations, but these demonstrate limited generalization [41, 42, 43]. Alternative approaches seek to derive action signals from video generation [44, 45, 46, 47, 48, 32, 49]. Affordances extracted from simulators [10, 50, 51, 52, 53] are another focus, yet they struggle with the significant sim-to-real domain-gap, particularly in 3D environments. Recent efforts [54, 11, 55, 56] attempt to directly acquire geometric-aware structured information in human video but require in-lab training or suffer from unstable performance. Instead, we leverage 3D flow-based affordance to achieve reliable zero-shot solutions.

**Keypoints and Flow for Robot Learning Systems** Previous studies use flow as action descriptors for robot learning [57, 58, 59, 60, 61]. However, these approaches either depend on embodiment-specific data, limiting their scalability [9, 7], or are simulation-based, facing significant sim-to-real domain-gap due to imperfect real-world RGBD point cloud generation [6, 8]. Wen et al. [62], Bharadhwaj et al. [63] utilizes flow as a prediction target for pretraining, but only operates in 2D flow and requires in-domain finetuning. In this paper, we extend the work of Seita et al. [7] into a more general version both in data resources and downstream applications. We acquire 3D flow prediction directly from RGBD human video datasets and achieve stable zero-shot skill transfer.

## 6 Conclusion

In this paper, we introduce General Flow as Foundation Affordance for scalable robot learning. For the first time, we develop a flow prediction model directly from large-scale RGBD human video datasets and successfully deploy it with a heuristic policy for stable zero-shot human-to-robot skill transfer. Our framework marks a stride in achieving scalability, wide application, and stable skill transfer concurrently. We believe our work paves the way for innovative research in scalable general robot learning. We comprehensively discuss the **limitations** of our framework in Appendix I.

## Acknowledgments

This work is supported by the Ministry of Science and Technology of the People´s Republic of China, the 2030 Innovation Megaprojects "Program on New Generation Artificial Intelligence" (Grant No. 2021AAA0150000). This work is also supported by the National Key R&D Program of China (2022ZD0161700).

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

<h1 style="text-align:center">Appendix</h1>

## A  Overview of Appendix

In this appendix, we offer additional implementation details and discussions of general flow. *The label pipeline, code and model weights will be released in the future*. Readers are also welcome to check out more details with them. This appendix is structured as follows:

- **Label Extraction:** we delve into the pipeline specifics utilized for extracting general flow labels from RGBD human video datasets in Appendix B.

- **Model Architecture and Training:** we provide an in-depth look at our model's architecture (Appendix C.1), augmentation techniques (Appendix C.2) and training process (Appendix C.3).

- **Baselines:** we provide more details of our baselines in Appendix D

- **Emergent Properties of General Flow:** We demonstrate the emergent properties of general flow from large-scale training in Appendix E.1. We also validate these properties via real-robot experiment in Appendix E.2.

- **Ablation Study:** A comprehensive quantitative ablation study is conducted in Appendix F, rigorously testing the effectiveness of our algorithmic design.

- **Real-World Experiment:** This section is dedicated to an expansive elucidation of real-world experiments, encompassing the experimental setup (Appendix G.1), real-world baseline comparisons (Appendix G.2), robot system development (Appendix G.3), policy derivation strategies (Appendix G.4), inference latency measurements (Appendix G.5), and analysis of failure cases (Appendix G.6).

- **Related Work of Real-world General Robot Learning:** we further discuss the related works of general robot policy training in Appendix H.

- **Limitations:** We comprehensively discuss the limitations of our current framework in Appendix I, including data diversity, policy backbone, manual grasping, representation format and heuristic policy.

- **Codebases:** Acknowledgements are extended to the multiple codebases that have been instrumental in supporting this project in Appendix J.

**Additional videos and flow visualizations** are available https://general-flow.github.io.

## B  Label Extraction Pipeline

General flow labels can be directly extracted from 3D human datasets or RGBD videos. Figure 6 displays some data resources we utilize.

### B.1  From HOI4D Datasets

We select the HOI4D dataset [15] as our primary resource due to its relatively large scale. This dataset offers comprehensive 3D labels, which are crucial for supporting 4D (point clouds + timestamps) Human-Object-Interaction (HOI) research. The labels we employ include RGBD images, camera parameters, object pose labels, scene segmentation masks, and action labels.

For effective closed-loop control, we divide the original action clips into multiple 1.5-second sub-clips, spaced at 0.15-second intervals, with a total of 3 time steps. For sub-clips that contain non-contact prefix actions (such as moving hands towards objects), we create 4 extended sub-clips with start timestamps within the semantic-less prefix.

The model's input comes from the first image of each sub-clip. We identify and match key elements (objects and hands involved in the manipulation) from the instructions with their corresponding

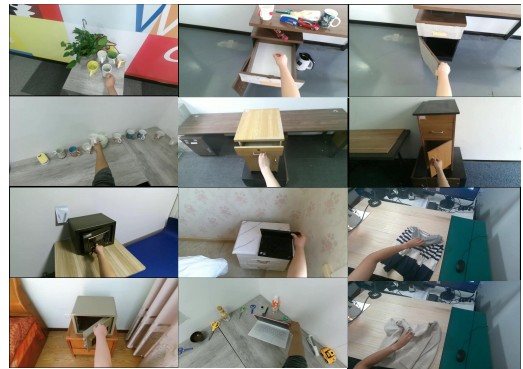

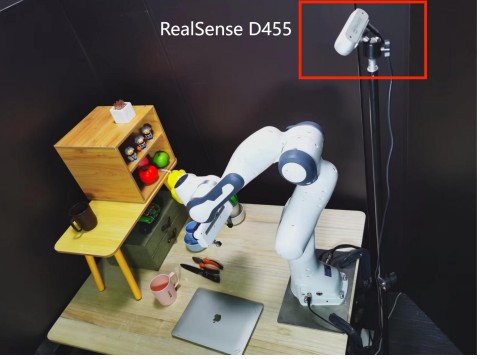

Figure 6: Examples of our cross-embodiment data resource.

Figure 7: Real-world deployment setting.

masks, considering the remainder as a background mask. Each mask is converted into a point cloud from RGBD values and down-sampled to one point per 0.02cm voxel. To adjust for noise in the HOI4D masks, we expand the hand mask by 8 pixels and shrink the object masks by 2 pixels.

We then proceed to extract general flow labels. Initial query points are selected within the masks of the objects of interest. Addressing segmentation noise, we maintain only the overlapping masks from the previous, current, and subsequent frames, using a homography matrix for projection onto the current frames. These points are chosen randomly, and their future trajectories are calculated based on ground-truth poses. We project all data back to the initial frame using the camera parameter labels. To correct for camera shake in the extrinsic parameter labels, we identify trajectories with shifts under 0.02 cm, compute their average as the camera shake, and deduct this from all points.

## B.2 From Collected RGBD Videos

Given the constraints of current RGBD Human-Object Interaction (HOI) datasets, which are either small in scale [34] or lack semantic richness [36] (mainly limited to pick & place actions), and considering the notable scarcity of resources for soft objects, we opt to collect our own RGBD videos. Our collection of RGBD videos for the "fold Clothes" task, captured with a D455 depth camera, includes 30 rollouts of 6 different types of clothing, resulting in 605 extracted clips. We plan to release this dataset in the future.

Echoing the process used in HOI4D, we maintain a duration of 1.5 seconds for each clip, with intervals of 0.15 seconds. Initially, we apply HOI segmentation [64] to acquire masks for hands and active objects. Utilizing HOI detection results from [16], we input bounding box outputs into FastSAM [17] for refined results. We retain only the masks with a confidence level above 0.5 for subsequent processing. After segmentation, we randomly sample 1024 points on the active object and employ co-tracker [18] to track their future positions in 2D pixel space. We exclude trajectories affected by occlusion, disappearance, or breakdown of depth values midway, and project the remaining trajectories back to 3D space and the first frame of the clip to derive the final general flow labels.

Our pipeline functions automatically, without the need for manual intervention. As general flow captures the geometric dynamics of the physical world, extending beyond mere object-centric interactions, our system effectively manages noise factors such as segmentation and point sampling errors (e.g., selecting query points on non-target objects due to segmentation mistakes), particularly during large-scale training.

# C  Model Architecture and Training

## C.1  Architecture

While the main body of our paper covers the bulk of our design, we offer further details in this section. The alignment width for CLIP [19] text features is set at 6, aligning with the dimension of the original point cloud features (RGB+XYZ). In the conditional Variational Autoencoder (VAE) [21] segment, we utilize a 2-layer Multilayer Perceptron (MLP) to encode the latent variable. This is followed by another 2-layer MLP that functions as the VAE decoder. We employ separate 2-layer MLPs for scale and normalized-trajectory prediction, each featuring a hidden dimension of 512. For ScaleFlow-B, our backbone configuration mirrors that of PointNeXt-B [20]. In ScaleFlow-L, we increase the backbone width from 32 to 64. Conversely, for ScaleFlow-S, PointNeXt-B is substituted with PointNeXt-S, and the CVAE utilizes a simpler 1-layer encoder and decoder, each with a hidden dimension of 384. The loss function parameters are configured as $\beta_1 = 25$ and $\beta_2, \beta_3 = 1$, with a focus on enhancing scale prediction. We maintain the latent variable dimension at 16. For more detailed information, please consult the configuration files in our code repository.

## C.2  Augmentation Techniques

In light of the complex environmental challenges encountered in zero-shot real-world deployments, we propose two technical augmentations to boost zero-shot generalization robustness. The fundamental concept involves simulating various states of hand occlusion and distributions of query points, compelling the model to adapt to diverse conditions:

- **Hand Mask (HM) Augmentation:** We encounter occlusions from human hands in our training data while facing occlusions from robot arms during deployments. Therefore, it is crucial to enhance the model's resilience to embodiment occlusions. To achieve this, we manipulate the presence of points on the hand in the input scene point clouds. We choose one of three rules, with probabilities $p_{h1}$=0.5, $p_{h2}$=0.2, and $p_{h3}$=0.3: (1) deleting all hand points; (2) keeping all hand points; and (3) sampling a random anchor point on the hand and retaining only points that are more than 12cm away from the anchor. These three rules are designed to simulate situations where the gripper experiences no occlusion, full occlusion, and partial occlusion during robot execution.

- **Query Points Sampling (QPS) Augmentation:** Different downstream applications may require varying query point sampling methods. Consequently, our model must be adaptable to various query point distributions. We achieve this by augmenting the training process. In each training iteration, we select a subset of available query points using one of two rules, based on probabilities $p_{s1}$=0.7, $p_{s2}$=0.3: (1) complete random sampling; (2) randomly selecting an anchor query point and then choosing a specific number of points closest to the anchor. The complete random sampling ensures that all points are covered, whereas anchor-based sampling maintains a structure more aligned with the format used in downstream action guidance generation.

## C.3  Training Details

For skills such as 'open safe', where most query points are static (e.g., points on the safe's body), direct model training leads to a strong bias towards predicting stationary trajectories. This results from a scale imbalance in our datasets. To mitigate this, we implement scale rebalance across the dataset. First, we employ the K-Means algorithm to cluster each data point's general flow by scale $L^i$. As a result, we obtain $N_r$ clusters of 3D points. We represent the original point ratios of each cluster as $\{r_i \mid i = 1..N_r\}$. Except for the cluster with the largest number of points, we perform resampling for all other clusters. The resampled distribution is given by:

$$\tilde{r}_i = \frac{e^{r_i/\tau}}{\sum_{i=1}^{N_r} e^{r_i/\tau}} \tag{3}$$

which is smoother than the original distribution. By default, we set $\tau$ to 1.

We utilize 1.5s video clips as our training data and set the time steps of general flow to 3 for all data sources. The dataset is divided into training, validation, and test sets in an 80%, 10%, 10% ratio, resulting in 51693, 6950, and 6835 clips respectively, with no identical object instances across sets. Each sample consists of 2048 scene points sampled in an $80\times80\times80\ cm^3$ cube space around the center of the flow start points using the furthest point sampling (FPS) algorithm (a 40 cm perception range suffices for most tasks). During training, we randomly sample 128 query points, while for validation, 512 points are randomly sampled. Standard augmentation techniques for point cloud prediction [20], including random rotation, shifting, scaling, coordinate normalization, color jittering, and feature dropping, are applied.

In each training iteration, we randomly select 128 trajectories from the available flow labels. Additionally, to boost robustness in downstream zero-shot prediction, we implement technical augmentations as described in the main paper. For scale rebalance, we set the number of clusters to 4 and the default temperature to 1. The training process utilizes the AdamW optimizer with a learning rate of 0.001 and a weight decay of 0.0001. We incorporate 10 warmup epochs, followed by a cosine scheduler for 200 epochs. The training for ScaleFlow-B can be completed within 10 hours using an Intel(R) Xeon(R) Gold 5220R CPU and a single NVIDIA GeForce RTX 3090 GPU.

Given that our datasets include ground-truth labels for object parts, we distribute 512 query points across each part equally during testing to enhance evaluation effectiveness. It's important to note that we do not use any part labels in model validation and real-word testing.

## D  Baselines

We adapt three types of relevant work to our setting:

- **2D Models:** To investigate the importance of 3D geometry information, we employ pretrained **ResNet** [23] and **Vision Transformer (VIT)** [24] models from the timm [65] library as feature extractors. We finetune these models, combining their 2D visual features with aligned text features and processing them through an MLP for direct flow regression. We also evaluate the performance of the **R3M representation** [25]. Both finetuning and frozen modes are considered for R3M.

- **VAT-MART [10]:** This model, originally designed for predicting affordance with single contact points, is adapted to our setting. We only utilize the 3D trajectory prediction branch of VAT-MART, replacing its task indentifier with aligned text features while keeping the rest of the model unchanged.

- **3D Backbones: FlowBot3D** [6] and **ToolFlowNet** [7] share a similar problem setting with ours. They originally used plain **PointNet++** [26] for flow prediction in simulation without language supervision. For fair comparison, we implement an improved version, replacing PointNet++ with the stronger **PointNeXt** [20] backbone as a geometric feature extractor. The extracted features, combined with aligned text features, are then processed through an MLP for general flow regression.

We preserve the architecture from the original repository, with the sole modification of incorporating text features before the prediction MLP to transition the model into a multimodal version. We derive aligned text features from the original CLIP features. The dimension of these text features is set to 32. For ResNet [23] and Vision Transformer [24], we utilize the standard 'ResNet18' and 'VIT-B-224' versions, respectively. The default pretrained weights are loaded using the Timm library [65]. For R3M [25], its 'ResNet18' version is employed. In our architecture, all MLPs dedicated to the final flow prediction consist of 2 layers, featuring hidden dimensions of 512 and 256, respectively. To ensure a fair comparison, scale rebalance, HMA augmentation and QPS augmentation are applied to all baselines.

| Model | ADE↓ | FDE↓ | ADE-H↓ | FDE-H↓ | Param (M) |
|---|---|---|---|---|---|
| ScaleFlow-S | 3.74 | 5.01 | 3.72 | 4.98 | 0.906 |
| ScaleFlow-B | 3.58 | 4.77 | 3.56 | 4.74 | 5.622 |
| ScaleFlow-L | 3.55 | 4.70 | 3.52 | 4.67 | 17.088 |

Table 3: Results of general flow prediction on the test set. "ADE-H" and "FDE-H" denote evaluations that include hand points in the model's input (with unit meter). With appropriate augmentations, our models are robust for hand occupancy.

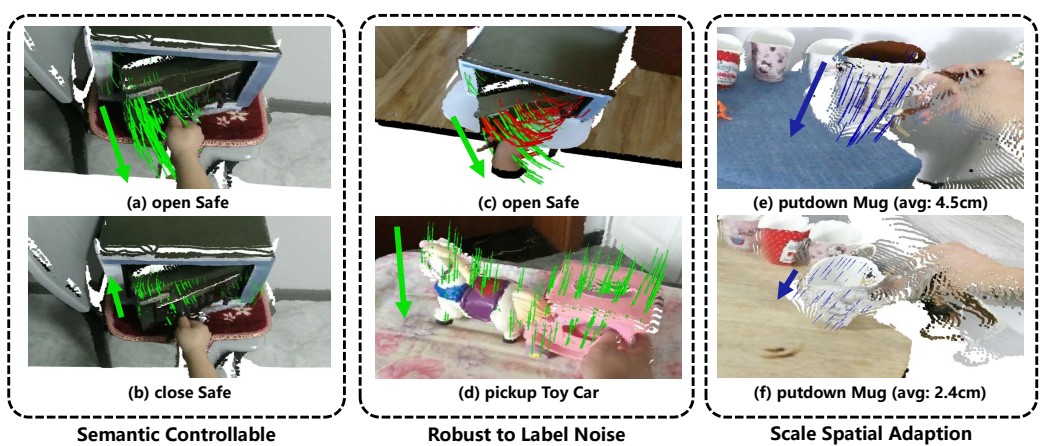

Figure 8: Emergent properties of general flow prediction are demonstrated. The arrow indicates the coarse direction of the predicted flow. In images (a) and (b), the same input is used, differing only in the text instruction. For (c) and (d), the color red represents the extracted label, while green denotes the model's prediction. In (e) and (f), "avg" signifies the average trajectory lengths of all query points.

# E  Emergent Properties of General Flow

## E.1  Properties Analysis

In this section, we demonstrate the notable emergent properties of general flow such as robustness to hand occlusion, language-driven semantic control, resilience to label noise, and scale spatial adaption.

**Robustness to hand occlusion**  We first test robustness to hand occupancy in inputs across all 3D models, which are denoted as ADE-H and FDE-H (evaluations that include hand points in the model's input). As shown in Table 3, with appropriate augmentations, our model are robust for hand occupancy.

**language-driven semantic control**  Through large-scale training, our model not onlyaptures rich semantic information but also becomes adeptly controllable through language modality. As depicted in Figure 8(a)(b), our model demonstrates the capability to predict varied flows for identical input point clouds when provided with different instructions.

**Resilience to label noise**  Furthermore, it is remarkably robust to label noise. Figure 8(c)(d) showcases two instances of this resilience: despite severe label noise (notable deviation in "open Safe" and near-static in "pickup Toy Car"), our model accurately predicts the correct trend.

**Scale spatial adaption**  Additionally, our model could perform scale spatial adaption through scalable training. It dynamically adjusts its prediction scale in response to the spatial relationships of objects, such as ending on the table and scaling up for longer distances, as seen in Figure 8(e)(f). All these emerging phenomena reveal the benefits of large-scale training.

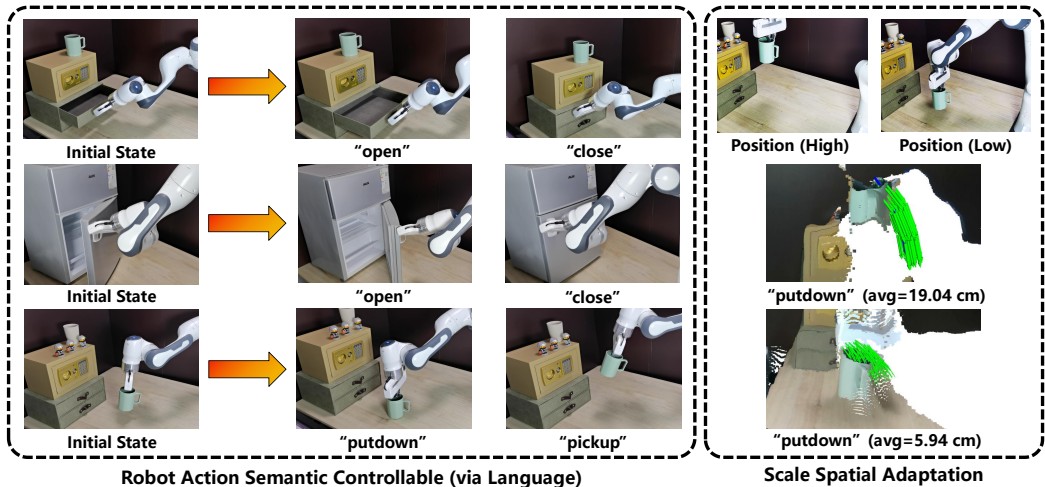

Figure 9: (Left) An experiment demonstrating action semantics control in a real robot through language instructions. By inputting various verbs of instruction, while keeping the initial states of the "Drawer", "Refrigerator", and "Mug" consistent, the robot is able to execute the correct semantic actions. (Right) A real robot demonstration showcasing the phenomenon of scale spatial adaptation. For the identical task of "putdown Mug", our model's predicted scale adjusts based on the Mug's position, averaging 19.04cm for higher positions and 5.94 cm for lower positions.

## E.2  Real-Robot Validation

We also validate the prediction properties through real-robot experiments. we confirm the "scale spatial adaptation" phenomenon in the "putdown Mug" task in a real-robot setting, as depicted to the right of Figure 9.

We then demonstrate the model's ability for language-controlled multi-semantic action generation. We select "Drawer," "Refrigerator," and "Mug" as manipulation objects, setting their initial states the same while varying only the instructional verbs for task execution. These results, presented to the left of Figure 9, illustrate that language instructions enable the execution of various behaviors within a single scene.

## F  Ablation Study

### F.1  Dataset Experiment Setting

We conduct an ablation study to evaluate the key design elements of our methods. The variants tested include:

- **w/o Text EarlyFusion:** aligned text features are concatenated with PointNeXt features (having a dimension of 32) instead of the original point clouds.
- **w/o Scale Normalization:** the Conditional Variational Auto Encoder (CVAE) predicts general flow without scale normalization. We explore two versions: one with absolute position prediction and another with relative displacement prediction.
- **w TDN Scale Normalization:** this approach employs normalization to adjust the length of absolute displacement to 1, rather than the total length.
- **w SDN Scale Normalization:** normalization is used to set the length of each step to 1.
- **w $\beta_1 = 1$ (weight of scale-loss):** this test is designed to assess the importance of adequately weighting scale prediction in the loss function.
- **w/o central crop:** all scene point clouds in a 2m operation space are fed into the model without cube space cropping. The active object points average only about 2% in this setup.

|  | Test-ADE (w/o hand) | Test-FDE (w/o hand) |
|---|---|---|
| Full | **3.58** | **4.77** |
| w/o Text EarlyFusion | 3.70 | 4.95 |
| w/o Scale Normalization (relative) | 3.76 | 5.04 |
| w/o Scale Normalization (absolute) | 3.81 | 5.12 |
| w/ TDN Scale Normalization | 3.74 | 5.00 |
| w/ SDN Scale Normalization | 3.77 | 5.10 |
| w/ $\beta_1 = 1$ (weight of scale-loss) | 3.68 | 4.93 |
| w/o central crop | 3.99 | 5.38 |
| w/o Scale Rebalance | 3.59 | 4.78 |
| w/o HMA Augmentation | 3.66 | 4.88 |
| w/o QPS Augmentation | **3.58** | **4.77** |

Table 4: Results of the ablation study on general flow prediction, with the best results highlighted in **bold** and the second-best results underlined. The unit of all metric is meter.

|  | open Safe | close Drawer | Avg |
|---|---|---|---|
| full | 9 / 10 | 10 / 10 | 95% |
| w/o Scale Rebalance | 7 / 10 | 8 / 10 | 75% |
| w/o HMA Augmentation | 8 / 10 | 6 / 10 | 70% |
| w/o QPS Augmentation | 5 / 10 | 4 / 10 | 45% |

Table 5: Result of the real-robot ablation study. We take the success rate as the evaluation metric.

- **w/o robustness augmentation:** these variants omit three types of technical augmentations (Scale Rebalance, Hand Mask Augmentation, query points sampling Augmentation) to determine their impact on the model's prediction accuracy in our benchmark.

The results of the ablation study are summarized in Table 4. For all variants except those without robustness augmentation, there is a noticeable degradation in model performance. In regards to the ablation of the three technical augmentations, it is evident that they do not detrimentally affect benchmark performance. Notably, the hand mask augmentation even significantly enhances in-domain prediction, which is an interesting observation.

## F.2 Real-Robot Ablation Study

We also investigate the role of designed augmentations in enhancing the robustness of zero-shot transfer via real-robot ablation study. We select two representative tasks ("open Safe" and "close Drawer") for evaluation. The ablation comparisons included:

- **w/o Scale Rebalance:** Utilizing the original flow label without rebalancing based on scale cluster results.

- **w/o HM Augmentation:** Setting $p_{h1} = 1.0$, which means erasing all points on the hand throughout the entire training process.

- **w/o QPS Augmentation:** Setting $p_{s1} = 1.0$, which means relying solely on random sampling for training query point selection.

The results in Table 5 indicate that each augmentation significantly contributes to robust zero-shot execution.

Notably, hand augmentation markedly affects tasks with substantial embodiment occupancy and occlusion, such as "close Drawer". Query point sampling augmentation emerges as particularly influential. Currently, the PointNeXt [20] architecture inherently couples the extraction of query point features, leading to a reliance on query point sampling augmentation in our framework. We anticipate that future advancements in disentanglement architecture within 3D learning will address this issue thoroughly.

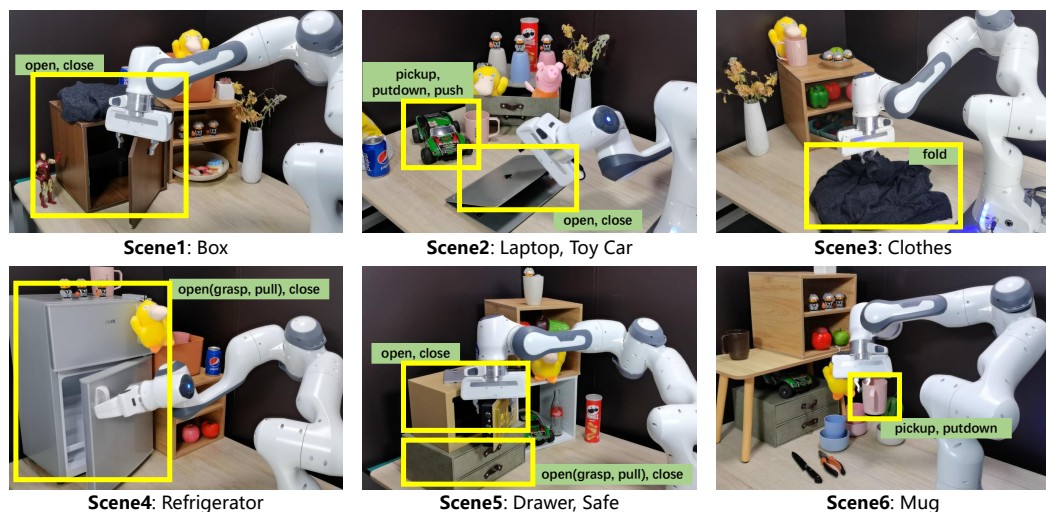

Figure 10: This figure illustrates the distribution of 8 objects across 18 tasks, encompassing various categories such as rigid, articulated, and soft bodies, arranged into 6 distinct scenes. Manipulated objects are highlighted within yellow bounding boxes, with each corresponding task denoted by a green box.

## G   Real-Robot Experiment

### G.1   Real-World Environment Setting

In our real-world experiment (Figure 10), we select 8 objects, including rigid, articulated, and soft bodies, as featured in our human video resources. We manually define multiple tasks and their corresponding success conditions for each object, resulting in a total of 18 distinct tasks. For a complete listing of these tasks, please refer to the content in the main paper (Section VI.B). For "Refrigerator" and "Drawer", we refer to them as "Storage Furniture" in the instructions. "Box" is also referred to as "Safe" since it can be seen as an approximation of an atypical design of "Safe".

These objects are arranged into 6 scenes, as depicted in Figure 10. For objects that are movable, such as "Mug" and "Toy Car", we randomly adjust their positions and orientations to add variability. It is worth noting that our experimental setup more accurately mirrors practical real-world scenarios compared to previous studies like Eisner et al. [6]. Our setup features diverse scenes and eliminates the necessity for clean object segmentation (left of Figure 11). This resemblance underscores the robustness and stability of our system in real-world scenarios.

The criteria for successful task completion varied. For "pickup" and "push" tasks, moving the object in the correct direction by more than 15cm is deemed successful. The "Putdown" action is deemed successful if the object is ultimately placed on the desktop and the orientation of the object is appropriate (for example, the mouth of a mug facing vertically upwards). For the "open" task of the revolute articulation structure, an opening of 80 degrees is considered a success. For its "close" task, bringing the object to within less than 5 degrees of the fully closed state is regarded as successful. 5 cm (to fully open or close state) is used as a criterion for prismatic structure. The "fold" is considered successful if one end of the garment reaches the other end.

### G.2   Real-World Baseline

To the best of our knowledge, Bahl et al. [11] is *the only open-source work* that shares a similar setting with ours, which involves learning a low-level affordance model directly from real-world human videos and could perform zero-shot prediction. We deploy this model in our experimental environment, using 'safe' and 'drawer' as text prompts for the grounding module [66]. Right of Figure 11 shows the visualization of the affordance prediction. Although it provides semantically

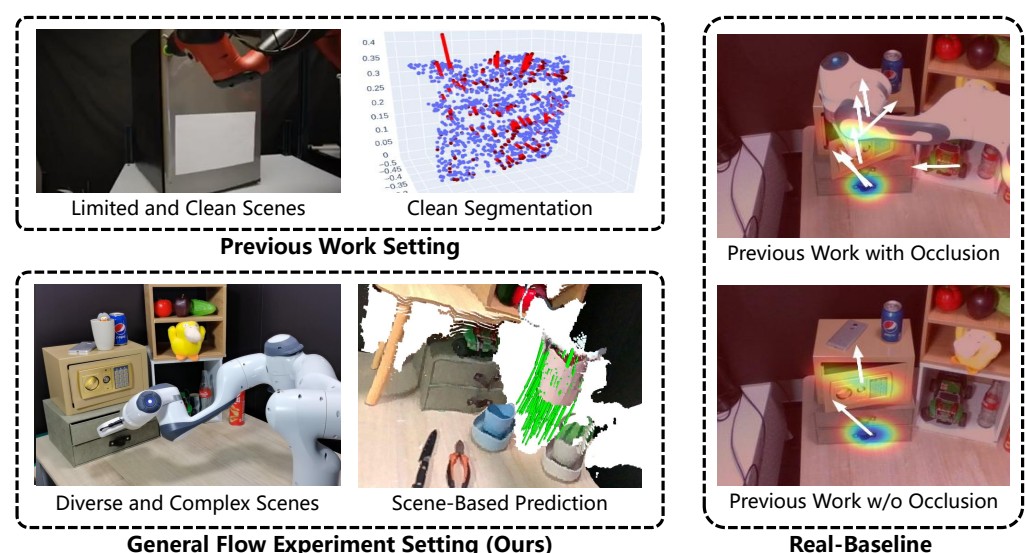

Figure 11: (Left) Our environment setting is much closer to practical real-world situations compared to previous work [6]. (Right) Affordance prediction from our baseline model [11] in the real world. The text prompts for the grounding module are set to "safe" and "drawer".

---

**Algorithm 1** Heuristic Close-Loop Policy from General Flow

---

**Require:** Task instruction $I$, camera stream $\mathcal{C}$, pretrained FastSAM model $\mathcal{M}_{seg}$, pretrained general flow predictor $\mathcal{M}_{flow}$, operation space controller $\mathcal{M}_{control}$.
    $p_{base} \leftarrow$ 2D position of Franka-Emika base
    $p_{extra} \leftarrow$ user interface (optional)
    **repeat**
        $O_{rgbd} \leftarrow \mathcal{C}$
        $O_{seg} \leftarrow \mathcal{M}_{seg}(O_{rgbd}, \text{prompts}=[p_{base}, p_{extra}])$
        Recover Point Clouds $P_{scene} \leftarrow BackProject(O_{seg})$
        Gripper Position $g \leftarrow \mathcal{M}_{control}$
        Query Points $Q \leftarrow Radius(P_{scene}, g, 10\text{cm})$
        General Flow $F \leftarrow \mathcal{M}_{flow}(P_{scene}, Q, I)$
        SE(3) Transformation $\mathcal{T} \leftarrow$ SVD-Alignment$(F)$
        Execution: $\mathcal{M}_{control}(\mathcal{T})$
    **until** Task Finished or Failed

---

meaningful predictions to some extent, it is limited by: (1) inadequate generalization for accurate motion direction prediction; (2) providing only 2D guidance without depth information; and (3) significant disturbances caused by embodiment occlusions. Given that the predicted post-grasp trajectories do not offer sufficient 3D guidance for closed-loop execution, we refrain from further robot execution trials.

### G.3 Development of the Robotic System

The heuristic policy based on close-loop flow prediction is demonstrated in Algorithm 1. Figure 7 shows a snapshot of our real-world deployment setup. We use a RealSense D455 RGBD camera to capture point cloud streams at a resolution of $1280 \times 720$, which is lower than the $1920 \times 1080$ resolution used in HOI4D [15]. Consequently, we opt for 0.01cm voxel downsampling during deployment, as opposed to the 0.02cm used in model training. The calibration parameters for our camera-robot system are (qw=0.911, qx=-0.015, qy=0.410, qz=-0.032) for orientation and (x=-0.265, y=0.260, z=1.095) for position. This configuration mimics a human ego-view manipulation perspective, beneficial not only for minimizing inference domain-gap but also aligning with practical applications in mobile robots. We utilize the RealSense camera's ROS driver for data acquisition.

The robot's base, static during manipulation, serves as a prompt for the FastSAM [17] model for robot segmentation. For enhanced accuracy, more prompt points or customized models [37] can be employed. Post-segmentation, we reconstruct 3D scene point clouds and select query points within 10cm of the gripper. These, along with the scene point clouds and the text instruction, are fed into our prediction model (ScaleFlow-B in our experiments) to obtain the anticipated general flow. We then apply the SVD algorithm[30] to obtain a robust transformation aligned with the predicted flow. The robot arm is driven by the Deoxys library [31] to follow the derived SE(3) transformation in a closed-loop manner, achieving a 0.4s inference latency (0.05s without FastSAM).

In practical applications, we note that the 6DoF controller for operational space in Deoxys lacks the necessary control precision for minute distances. Consequently, for trajectories shorter than 5 cm, we adopt a strategy of consolidating all steps into one and scaling this unified step to 5 cm in length. This method significantly enhances control accuracy over shorter distances, boosting the system's overall effectiveness and efficiency. Approximately 25% of predictions activate this workaround, which is considered acceptable given that the majority of tasks do not demand high levels of dexterity. Future improvements could include more precise controllers and calibration. The loop rate for our ROS system is set at 20 Hz. For safety, we manually confirm each planning step, although we find this almost unnecessary, as all experiments proceed with continuous pressing and confirmation without any delays.

### G.4 Heuristic Policy Derivation

This section offers an in-depth explanation of our heuristic policy derivation. We begin by obtaining the gripper pose from the Deoxys API and projecting it into the camera's coordinates using calibration parameters. We select points within a 10cm distance from the gripper. Utilizing these points, we predict general flow and proceed to derive a 6DoF end-effector motion plan in camera space. For point clouds $k_t$ and $k_{t+1}$, each containing $N$ points and representing adjacent timestamps, our objective is to identify a 6DoF transformation with rotation $\hat{R}$ and translation $\hat{T}$ that fulfills the following criteria:

$$
w_i = \left( \frac{\frac{1}{d_i+\beta}}{\sum_{j=1}^{N} \frac{1}{d_j+\beta}} \right)
$$
$$
\hat{R}, \hat{T} = \arg\min_{R,T} w_i \left\| k_{t+1}^i - (R \cdot k_t^i + T) \right\|^2
\tag{4}
$$

where $w_i$ denotes the regression weight inversely proportional to the distance $d_i$ between the $i$-th query points and the gripper position, with $\beta$ set to 1. We solve Equation 4 using the SVD algorithm [30] for robust results. The acquired transformation $\mathcal{T} = (\hat{R}, \hat{T})$ is then projected back into the robot's coordinates and adjusted to the gripper's coordinates for controller execution.

### G.5 Inference Latency

Table 6 details the average inference latency of each component in our pipeline, based on 10 measurements of the "open(grasp) Refrigerator" task. The significant bottleneck is the FastSAM segmentation, which contributes to 85.7% of the latency. This highlights the need for more efficient open-world segmentation models in future work. Without FastSAM segmentation, general flow prediction is not the sole system bottleneck; stream acquisition of point clouds also presents substantial room for improvement.

### G.6 Failure Case Analysis

We analyze failure cases for tasks with success rates below 60% and suggest potential improvement methods:

| Part | Time (ms) |
|------|-----------|
| Data Acquisition | 3.2 |
| FastSAM Segmentation | 347.6 |
| PointCloud Generation | 30.8 |
| Query Points Sampling | 0.3 |
| Flow Prediction (ScaleFlow-B) | 22.1 |
| Heuristic Policy Generation | 1.7 |
| Total (with Segment) | 405.7 (2.5Hz) |
| Total (without Segment) | 58.1 (17.5Hz) |

Table 6: The inference latency of each part in our pipeline. The results are the average values of 10 measurements.

- *"Push Toy Car" (50% success rate)*: The Toy Car's direction requires a sophisticated semantic understanding. To improve our model's capability in this area, additional data collection is essential. Due to the toy car's small size, integrating a wrist camera could also help mitigate significant occlusion problems and enhance performance.

- *"Open Drawer" (30% grasping, 40% pulling)*: The mixture of prismatic and revolute structures in the HOI4D [15] datasets leads to a slight tendency towards including rotational components in predictions. Its negative impact is amplified in our fabric cabinet with high friction. The leather handle on the drawer also poses a challenge, often slipping from the gripper. Future enhancements could include using larger datasets with more robust language semantics [67] and redesigning the gripper.

- *"Open Laptop" (50%)*: The laptop's thin lid often results in poor or incorrect RGBD point cloud generation. Utilizing point clouds fused from multiple camera views could ameliorate this issue.

Failure case videos are available in https://general-flow.github.io. In conclusion, most of these problems are addressable in future deployments. We systematically summarize these limitations and potential improvements in the next section.

## H  Related Works of Real-World General Robot Learning

Research in general-purpose robotic manipulation in real-world settings is constantly evolving, with a focus on integrating Large Language Models (LLMs) for high-level planning [68, 69, 70, 71, 72, 73] and exploring direct actionable guidance through LLMs [74], albeit with challenges due to overlooked physics dynamics. The development of large models for direct low-level control [75, 76, 77, 78] faces scalability issues due to intensive data requirements [3, 4]. This highlights the need for a training framework that achieves a balance between actionable output and scalability. In this work, we achieve this through a scalable robot learning approach based on general flow prediction.

## I  Limitations

The diversity and volume of our data are currently limited, which restricts our model's ability to provide adequate guidance for complex tasks. A future direction could involve utilizing larger RGBD datasets [79] or RGB datasets [35, 80] combined with depth estimation techniques [81, 32] to address these issues.

We currently employ a Conditional VAE to capture the multimodal distribution of human behavior in our dataset. However, this architecture may lack sufficient expressive power when dealing with more challenging scenarios, such as larger and more diverse datasets. To address this limitation, the Diffusion model [82] could be a potential solution.

In addition, the manual grasping process could be replaced with heuristic policies such as those described in AVDC[32], or through more advanced and automated methods like DexFunc[14], MoKa[13], and ManipVQA[83] in the future. It is also feasible to expand the representation from object-centric affordance to universal keypoints motion, similar to what is described in ATM[62].

The initially adopted ICP-based heuristic policy in a zero-shot manner might limit the completion of tasks, particularly those requiring contact-rich manipulation. Future studies could fine-tune policies based on general flow predictions within a few-shot imitation learning setting, such as Track2Act[63], or use general flow as an action constraint [29] for sample-efficient reinforcement learning, akin to VRB [11].

## J  Codebases

We extend our gratitude to the following codebases for their support in the development of this work:

- The model training framework and the 3D backbone are based on the codebase from Qian et al. [20].
- For HOI4D point-cloud data processing, we adopt methods from Liu et al. [15].
- For Hand-Object-Interaction (HOI) detection, we utilize the 100DOH tools from Shan et al. [16].
- FastSAM (Zhao et al. [17]) is used for all segmentation in this work.
- The co-tracker from Karaev et al. [18] is employed to track points in pixel space.
- Implementations of ResNet, Vision Transformer, R3M, and the VAT-MART baseline are adopted from Bao et al. [27], Nair et al. [25], and Wu et al. [10].
- The ros-perception and data stream acquisition are based on Shridhar et al. [84].
- We inherit the SVD transformation solver directly from Zhong et al. [85].
- The impedance controller for the end-effector is adopted from the Doexys library (Zhu et al. [31]).

