# OpenReview forum: "General Flow as Foundation Affordance for Scalable Robot Learning"
_robot-learning.org/CoRL/2024/Conference — CoRL 2024_

### Official Review · Reviewer_Y7U2 · 2024-07-20
**A well-motivated idea and a clean method, but need to rejudge its significance.**

**Originality:** 3
**Technical Quality:** 3
**Clarity Of Presentation:** 4
**Potential Impact:** 3
**Recommendation:** 3
**Confidence:** 4

**Review:**

The paper presents good motivation and intuition of why predicting the 3D flows benefits zero-shot human-to-robot skill transfer. The flow learning pipeline is clear and simple enough. The scale-aware prediction target and the augmentation techniques are natural and make sense. The authors have done adequate experiments to demonstrate that the model can predict better 3D flows in different scenarios than baseline architectures. The real world manipulation experiments show that the predicted flow can be easily mapped to a downstream policy.

My main concern is the "generalist" and "foundation" of the 3D flow in robot manipulation, as is emphasized by the authors.
1. The 3D flows are a coarse estimate of object affordance and inevitably comes with noise and inconsistency, which makes it not a suit for tasks requiring higher accuracy like contact-rich manipulation (e.g. insertion). For articulated objects, a policy based on 3D flows may break the object because 3D flows do not reveal the articulated property.
2. There is no natural way of getting the query points automatically. In the real world experiments, the authors manually select these points near the gripper, which largely affects its generalization ability.
3. To use the predicted flows, the authors assume that the gripper is already in a good manipulation state to the robot (i.e. grasping), making it harder to directly apply to common tasks. Also, how to manipulate an object can depend on how the object is being grasped, but the predicted flow does not consider the grasping state (https://sites.google.com/view/task-oriented-grasp/).
4. It is unclear how robust the flows are to the change of the camera views.
5. It is unclear if the 3D flows can be used in tasks requiring obstacle avoidance (https://interactive-language.github.io/).
6. The authors utilize a VAE to capture the multimodality property, but there is no experiment showing that the model learns it. Can the model predict all possible flows in such tasks like "pushing T" (https://diffusion-policy.cs.columbia.edu/)?
7. The authors mention ATM (https://xingyu-lin.github.io/atm/) in the limitation section, which has a similar idea of predicting flows. More discussions on the difference is necessary, preferably with experiments. The authors need to clarify the benefits of predicting a 3D flow than a 2D one, and the potential limitation of the former compared with the later.

**Quality Of The Limitations Section:**

2

**Questions For Rebuttal:**

Please see the bullet points in Review. The authors should proof the generalist of the method in various scenarios, or limit their claims to the extent that the method actually applies to.

**Robotics Focus:**

4

**Summary Of Paper:**

The paper aims to increase the scalability of zero-shot human-to-robot skill transfer in real-world language-conditioned manipulation tasks. It proposes to predict the 3D flow of RGBD videos as a general guidance for the downstream manipulation policy. To achieve robust flow prediction, scale-aware strategies and augmentation techniques are employed. Through experiments, the method demonstrates more accurate flow predictions over baseline models and high success rate in zero-shot human-to-robot skill transfer in real world manipulation tasks.

**Summary Of Recommendation:**

I list the paper as weak accept because the idea is well-motivated and the method is clear and works in enough scenarios. However, the experiments are not sufficient to show whether the 3D flows are general and foundational, which requires more work. I will raise my score if the authors are able to address large part of my concerns.

---

### Official Review · Reviewer_6L29 · 2024-07-23
**Learning a general object flow prediction model**

**Originality:** 2
**Technical Quality:** 2
**Clarity Of Presentation:** 3
**Potential Impact:** 2
**Recommendation:** 3
**Confidence:** 4

**Review:**

The authors propose a method to learn a language-conditioned object flow prediction model, called ScaleFlow. The authors claim this model could be benefitial to learn robot skills directly from video.

To achieve this goal, the authors:
- Generate a labeled dataset, combining available online datasets (HOI4D) and collected RGBD videos.
- Train a language-conditioned flow prediction network. To do so, they combine CLIP features with PointNeXt from ToolFlownet.
- Design a controller to generate robot actions given the predicted object's flow.

The paper is reasonable easy to read and they motivate their goal of learning a foundation model of object's flow as useful representation to scale from videos.

**Strenghts**
- The ultimate goal of building an object's flow prediction model is sound for robotics as a middle way to learn robot skills from videos.
- The authors show impressive zero-shot transfer. This is particularly impressive given the small size of the used dataset.


**Weaknesses**
- It is hard to clearly find what are the novel contributions in the work and what is not. For example, ToolFlowNet already predicts the flow of the objects. The paper could benefit from clearly stating in the introduction on the shoulders of which previous work they are building their method. For example, they could reframe some paragraphs in the introduction stating what the community has been able to do until now and what is missing that they propose. Without this clarification, it is hard to measure the real contributions of the work.
- Also, related with the literature, previous works, such as ScrewNet [1] or FlowBot 3D also propose a category-free, general object motion prediction model. The authors could also clarify how their method differs from these previous approaches and what are the benefits of their approach with respect to them.
- The authors should clearly specify if the zero-shot transfer is happening in arbitrary novel objects or in objects belonging to the training set. If they test in only train, it would be nice to provide a table comparison on success rate on object's out of the training dataset.

In summary, despite the task of learning a foundation model for general object motion prediction is a highly relevant one, the paper should focus on:
1. Cleary state how their general method is different from previous approaches.
2. Present an extensive evaluation on objects in-demonstrations and out-of-demonstrations to show the generalizable capabilities of the presented model.

[1] Jain, Ajinkya, et al. "Screwnet: Category-independent articulation model estimation from depth images using screw theory." 2021 IEEE International Conference on Robotics and Automation (ICRA). IEEE, 2021.

**Quality Of The Limitations Section:**

1

**Questions For Rebuttal:**

- Beyond the big picture of building a foundational model for object flow's, what are the technical contributions of the work?
- The Zero-Shot transfer is pretty impressive. Did you test it in object's that were not in the training dataset?
- If not, how well does the model perform when you try to generate object's flows for objects that were not in the dataset? If the model does not adapt to novel objects, calling the model a foundation model might be an overclaim, and the paper would benefit from reducing the overclaims.

**Robotics Focus:**

4

**Summary Of Paper:**

The paper introduces a method to learn a language conditioned object flow prediction model from RGBD data. This model is afterwards exploited for robot action generation. The authors claim an 81% success rate in Zero-Shot transfer.

**Summary Of Recommendation:**

After the rebuttal period, the authors correctly answered my concernes. Thus, I have updated my score to weak accept.

---

### Official Review · Reviewer_7196 · 2024-07-24

**Originality:** 5
**Technical Quality:** 3
**Clarity Of Presentation:** 3
**Potential Impact:** 4
**Recommendation:** 4
**Confidence:** 4

**Review:**

This paper presents a novel approach to robot learning by using RGBD human video data to train a language-conditioned 3D flow prediction model. The proposed method, which predicts future trajectories of 3D points, demonstrates impressive zero-shot transfer to robots, and provides promising results with real-world tasks. The paper's analysis provides strong evidence for the method's performance and robustness. However, the requirement for grasp initialization, complexity of the architecture, and reliance on RGBD data are significant drawbacks that may limit scalability.

**Strengths**

- **Potential for Scalability:** The framework utilizes actionless RGBD human video data, potentially addressing the challenge of data scalability in robot learning.

- **Generalization:** The model demonstrates zero-shot skill transfer across multiple tasks and scenes.

- **Robustness:** The use of augmentations such as hand mask augmentation and query point sampling augmentation improves the model's robustness to occlusions and diverse conditions.

- **Comprehensive Evaluation:** The paper includes thorough experimental evaluations and ablation studies, providing detailed insights into the system's performance and properties.

**Weaknesses**

- **Grasp Initialization Requirement:** The need to initialize with a grasp limits the range of possible tasks and scenarios, as evidenced by the lower success rate in tasks like pushing.

- **Complexity:** The architecture is complex and not fully justified by the ablation studies, leaving room to question the necessity of all components.

- **Inference Speed:** Although it is addressed in the appendix, the inference speed, particularly the latency due to segmentation, could be a bottleneck in real-time applications.

- **Scalability Limitation:** The reliance on RGBD data, although mentioned as a limitation by the authors, does restrict the scalability to environments where such data can be consistently captured.

**Quality Of The Limitations Section:**

3

**Questions For Rebuttal:**

- Can you expand on your claim in Section 3.2 that using a VAE helps capture multimodal distributions, considering that a Gaussian prior and regression loss typically assume a unimodal distribution? For example, given the same point cloud and task description, if the flow distribution is multimodal, would you not observe averaging behavior between these modes in your predictions?

- Please clarify the generalization capabilities of your model. Are all object categories and tasks in evaluation seen in HOI4D or your custom dataset? Generalization from "Safe" to "Box" does not seem sufficient to claim generalization to novel object categories, as is claimed in Section 4.3.

- Could this method be extended to enable grasp prediction by adding query points not only to objects but also to the agent itself? Or does this expand the domain gap too much for zero-shot generalization?

**Robotics Focus:**

4

**Summary Of Paper:**

The paper presents a method for robotic manipulation using a language-conditioned 3D flow prediction model. The key idea is the use of 3D flow, which represents future trajectories of 3D points on task-relevant objects. The model is trained on relatively large-scale RGBD human video datasets and achieves zero-shot skill transfer to robots, with an 81% success rate across 18 tasks in 6 scenes. The approach leverages the scalability of human video data and incorporates several methods of enhancing robustness and stability.

**Summary Of Recommendation:**

The paper introduces a promising approach to robot learning through the use of 3D flow prediction, enabling learning from actionless videos, a potentially impactful contribution to the field. The authors acknowledge several limitations, but some concerns remain. The initial results are promising, yet the dataset is relatively small, limiting the model's generalization capabilities. Additionally, the current heuristic method for transferring predicted flow to policies requires manual grasp initialization, which constrains task diversity. Scalability concerns exist due to the reliance on RGBD data, though future work incorporating monocular depth estimation methods may address these issues. Despite these challenges, the paper makes a significant contribution and deserves to be accepted, with several potential opportunities for further improvement.

---

### Author Rebuttal · Authors · 2024-08-07

The revised manuscript

---

### Decision · Program_Chairs · 2024-09-04

**Decision:**

Accept

**Comment:**

This paper proposes using 3D flow as a more general affordance representation for scalable robot learning. All reviewers recognize the importance and relevance of this work and are generally positive about the experiments and analysis presented.

Before the rebuttal, a major concern among the reviewers was whether the method could genuinely be considered "foundational." Although the definition of a foundation model may vary, the reviewers suggested that a more diversified and in-depth evaluation across a larger set of novel scenes might be necessary. Additional concerns included the need for manual grasp initialization, reliance on RGBD data, and the lack of detailed comparisons with previous methods.

During the rebuttal, the authors did an excellent job addressing the reviewers’ concerns by providing further clarifications on their method’s generalization capabilities and better positioning their work within the existing literature. The final ratings were one Strong Accept, one Weak Accept, and one Weak Reject. The reviewer who rated the paper as Weak Reject did not engage in the discussion during or after the rebuttal. From my perspective, the authors provided sufficient clarifications to address this reviewer’s concerns.

Overall, this is an interesting paper that offers a new perspective on affordance representation. However, the authors should carefully revise the paper to ensure it is fairly positioned within the broader literature.